# Melanoma Cell State-Specific Responses to TNFα

**DOI:** 10.3390/biomedicines9060605

**Published:** 2021-05-26

**Authors:** Su Yin Lim, Sara Alavi, Zizhen Ming, Elena Shklovskaya, Carina Fung, Ashleigh Stewart, Helen Rizos

**Affiliations:** 1Department of Biomedical Sciences, Faculty of Medicine, Health and Human Sciences, Macquarie University, Sydney, NSW 2109, Australia; esther.lim@mq.edu.au (S.Y.L.); jen.ming@mq.edu.au (Z.M.); elena.shklovskaya@mq.edu.au (E.S.); carina.lauter@mq.edu.au (C.F.); ashleigh.stewart@mq.edu.au (A.S.); 2Melanoma Institute Australia, Sydney, NSW 2065, Australia; s.alavi@centenary.org.au; 3Melanoma Oncology and Immunology, Centenary Institute, Camperdown, NSW 2050, Australia

**Keywords:** dedifferentiation, immunotherapy, antigen presentation, immune checkpoint inhibitors

## Abstract

Immune checkpoint inhibitors that target the programmed cell death protein 1 (PD1) pathway have revolutionized the treatment of patients with advanced metastatic melanoma. PD1 inhibitors reinvigorate exhausted tumor-reactive T cells, thus restoring anti-tumor immunity. Tumor necrosis factor alpha (TNFα) is abundantly expressed as a consequence of T cell activation and can have pleiotropic effects on melanoma response and resistance to PD1 inhibitors. In this study, we examined the influence of TNFα on markers of melanoma dedifferentiation, antigen presentation and immune inhibition in a panel of 40 melanoma cell lines. We report that TNFα signaling is retained in all melanomas but the downstream impact of TNFα was dependent on the differentiation status of melanoma cells. We show that TNFα is a poor inducer of antigen presentation molecules HLA-ABC and HLA-DR but readily induces the PD-L2 immune checkpoint in melanoma cells. Our results suggest that TNFα promotes dynamic changes in melanoma cells that may favor immunotherapy resistance.

## 1. Introduction

Immune checkpoint inhibitors targeting the programmed death protein 1 (PD1) immune checkpoint protein have significantly improved outcomes of patients with melanoma. Updated phase III clinical trial data (e.g., CheckMate 067, KEYNOTE-006) report objective response rates of 45% and 5-year overall survival of 39–44% among patients with advanced melanoma [1]. PD1 blockade reinvigorates effector CD8 T cells [2] and these tumor-reactive T cells secrete cytokines, including interferon gamma and tumor necrosis factor alpha (TNFα) and cytolytic enzymes, to kill tumor cells. Although TNFα can act as an effector molecule for cytotoxic T cells [3], in the context of PD1 blockade, this cytokine may contribute to immune resistance. For instance, TNFα diminishes the survival of CD8 T cells [4] and induces the expression of immune suppressive molecules such as PD-L1 [5] and CD73 [6].

TNFα also promotes melanoma dedifferentiation by suppressing the expression of the microphthalmia-associated transcription factor (MITF), a master regulator of the pigmentation pathway and melanocytic antigens, including the *TRP1*, *DCT* and *MLANA* genes [7]. The loss of these wild-type antigens decreases melanoma immunogenicity and favors immune escape [8,9]. The gradual reduction in MITF occurs along with TNFα-induced expression of the neural-crest marker NGFR [10,11]. NGFR^hi^ melanoma cells pre-exist in all melanoma tumors [12], are intrinsically resistant to targeted and immune checkpoint inhibitor therapies and are selectively expanded during treatment [9,13]. The immune suppressive effects of TNFα have led to the TICIMEL phase 1b clinical trial evaluating the activity and safety of combination anti-TNFα and immune checkpoint inhibitors (anti-PD1 and anti-CTLA4) in advanced melanoma (NCT03293784).

In this study, we examined the pleiotropic effects of TNFα on 40 melanoma cell lines. The influence of exogenous TNFα on melanoma dedifferentiation markers, immune inhibitory checkpoints PD-L1 and PD-L2, and antigen-presenting molecules HLA-ABC and HLA-DR was examined. We found that although TNFα signaling was intact in all melanoma cells, the response to TNFα was dependent on the differentiation status of melanoma cells, with the melanocytic and transitory melanomas showing more pronounced downregulation of the pigmentation regulator MITF. Furthermore, we demonstrate that TNFα is a poor inducer of antigen presentation molecules HLA-ABC and HLA-DR in melanoma cells. Our results suggest that TNFα produces a dynamic program of transcriptome and protein changes that may favor immunotherapy resistance in melanoma. These changes include the loss of wild type antigens, and the induction of the PD-L2 immune checkpoint with the minimal induction of antigen presentation molecules.

## 2. Materials and Methods

### 2.1. Cell Lines and Culture

A total of 40 melanoma cell lines were included in this study, provided by Prof. Nicholas Hayward and Prof. Peter Parsons at QIMR Berghofer Medical Research Institute (Australia), Prof. Bruce Ksander at Harvard Medical School (United States), Prof. Peter Hersey at the Centenary Institute (Australia) and Prof. Xu Dong Zhang at the University of Newcastle (Australia). The oncogenic driver mutation status of these cell lines has been previously reported [14]. Two short-term melanoma cell lines were cultured from surgically excised, enzymatically processed melanoma lesions (SCC14-0257, SMU15-0217) in a study carried out in accordance with the recommendations of Human Research ethics committee protocols from Royal Prince Alfred Hospital (Protocol X15-0454 and HREC/11/RPAH/444) [14]. Cell authentication was confirmed using the StemElite ID system from Promega (Madison, WI, USA).

Cell lines were cultured in Dulbecco’s Modified Eagle Medium (DMEM) or Roswell Park Memorial Institute-1640 media (RPMI) supplemented with 10 or 20% heat-inactivated fetal bovine serum (FBS; Sigma-Aldrich, St. Louis, MO, USA), 4 mM glutamine (Gibco, Thermo Fisher Scientific, Waltham, MA, USA) and 20 mM HEPES (Gibco) and were maintained at 37 °C in 5% CO_2_. For TNFα treatment, melanoma cells were seeded in 6-well plates (0.5–1 × 10^5^ cells/well) or in T75 flasks (0.5–1 × 10^6^ cells/flask), and after an overnight incubation, cells were treated for 72 h with 1000 U/mL TNFα (Peprotech, Rocky Hill, NJ, USA) or control (0.1% bovine serum albumin (BSA, Sigma-Aldrich) in phosphate-buffered saline (PBS, Gibco).

### 2.2. Flow Cytometry

Staining was performed in PBS supplemented with 5% FBS, 10 mM EDTA and 0.05% sodium azide. Cells (0.3–0.5 × 10^5^) were incubated for 30 min on ice with mouse anti-human antibodies against HLA-ABC (1:100, clone W6/32, Cat no: 311438), HLA-DR (1:100, clone L243, Cat no: 740302), CD271/NGFR (1:50, clone ME20.4, Cat no: 345109), CD273/PD-L2 (1:50, clone 24F.10C12, Cat no: 329628) (all from BioLegend, San Diego, CA, USA) and CD274/PD-L1 (1:40, clone MIH1, Cat no: 563738; BD Biosciences, Franklin Lakes, NJ, USA) conjugated to phycoerythrin (PE), fluorescein isothiocyanate (FITC), PE-cyanine 7 (PE-Cy7), allophycocyanin (APC) and brilliant violet 421 (BV421), respectively. All antibodies were titrated prior to use to ensure optimal concentrations were selected. Fc block (BD Biosciences) was used to prevent non-specific staining due to antibody binding to Fc receptors. Fluorescence minus one controls (FMO, staining with all but one antibody for each fluorochrome) were included with each experiment. Cell viability was determined by staining cells with either 5 µM of DAPI (Invitrogen, Thermo Fisher Scientific), Zombie Yellow dye (Biolegend) or Live Dead near-IR fixable dye (Invitrogen, Thermo Fisher Scientific).

Samples were acquired on the BD LSRFortessa X20 flow cytometer (BD Biosciences) and the FlowJo software (TreeStar, Ashland, OR, USA) was used for data analysis. At least 10,000 live events were acquired. The general gating strategy included a forward and side scatter area (FSC-A and SSC-A) to exclude cell debris, a time parameter to exclude electronic noise, a forward scatter-area and height (FSC-A/FSC-H) to exclude doublets and gating on viable cells (by gating on DAPI, Zombie Yellow or Live Dead near-IR negative events). Relative marker expression levels were calculated by dividing the geometric mean fluorescence intensity (MFI) of the antibody-stained sample by the FMO control MFI. Relative MFI is used in all analyses, and a relative MFI > 1.5 was considered to reflect positive expression relative to the control.

### 2.3. Western Blotting

Total cellular proteins were extracted and resolved on 10% SDS–polyacrylamide gels before transferring to Immobilon-FL PVDF membranes (Millipore, Bedford, MA, USA) as previously described [15]. Western blots were probed with the following primary antibodies targeting: MITF (1:1000, C5, Cat no: OP126L; Calbiochem, San Diego, CA, USA), AXL (1:200, Cat no: AF154; R&D Systems, Minneapolis, MN, USA,), Melan A (MART-1, 1:1000, Cat no: 34511; Cell Signaling, Danvers, MA, USA), total IκB (1:1000, Cat no: 4812; Cell Signaling), and ß-actin (1:6000, Cat no: A5316; Sigma-Aldrich) overnight at 4 °C. Membranes were detected on the ChemiDoc MP imaging system (BioRad, Hercules, CA, USA) and densitometry quantification performed using the Image Lab software (BioRad).

Densitometric values for each protein were normalized to ß-actin and values were converted to z-scores to enable the analysis of independent Western blotting experiments.

### 2.4. RNA Sequencing and Transcriptome Analysis

Total RNA was isolated from melanoma cells using the RNeasy Kit (Qiagen, Hilden, Germany). cDNA synthesis and library construction were performed using the TruSeq RNA Library Prep Kit (Illumina, San Diego, CA, USA) and paired-end 100 bp sequencing, with each sample yielding 40–50 million reads. Sequencing was performed on the Illumina NovaSeq platform at the Australian Genome Research Facility. RNA data processing was performed as previously described [16]. Briefly, the trimming of Illumina TruSeq was carried out using cutadapt and filtered reads were mapped to reference genome hg38. Reads were imported into R with GenomicAligments read GAlignmentPairs function and GENCODE Genes version 26 was used as the gene reference database.

RNA counts were normalized using the weighted trimmed mean of M-values (TMM) implemented in the edgeR Bioconductor package. Normalized counts were transformed using *voom*, as implemented in the *Limma* package [17]. For single-sample gene set enrichment analysis (ssGSEA), RSEM was used to derive FPKM estimates using GENCODE Genes version 26 as the reference database. Absolute signature enrichment scores were determined using ssGSEA version 9.1.1 [18] provided by Gene Pattern [19] with the Hallmark gene sets of the Molecular Signature Database version 6.2 [20] and the melanoma subtype gene sets [21]. A false discovery (FDR) adjusted *p* value (*q* value) < 0.1 was used for comparisons of gene signatures between two groups using the t test within the Morpheus web-based tool (https://clue.io/morpheus, accessed on 1 February 2021).

### 2.5. Statistical Analysis

Statistical analysis was performed using the GraphPad Prism software version 9 (GraphPad, San Diego, CA, USA). All values are expressed as the mean of at least three independent experiments and statistical methods applied are detailed in each figure legend. Differences were considered to be statistically significant when *p* < 0.05.

## 3. Results

### 3.1. TNFα Pathway Signaling Is Ubiquitous in All Melanomas

We initially examined TNFα signaling in response to exogenous TNFα in 40 melanoma cell lines, including 31 cutaneous (11 BRAF-mutant, 10 NRAS-mutant, 10 BRAF/RAS WT) and nine uveal melanoma cell lines (five GNAQ-mutant, three GNA11-mutant and one GNAQ/GNA11 WT). To analyze melanoma cell responses to TNFα, we evaluated the accumulation of IκB, an NF-κB inhibitor which is rapidly phosphorylated and degraded in response to TNFα. All 40 melanoma cell lines in our panel responded to 1000 U/mL TNFα with diminished IκB protein accumulation (Figure 1A), and although the degree of IκB reduction was variable (one to six-fold reduction in the presence of TNFα), all 40 melanoma cell lines showed reduced IκB accumulation (Figure 1B). There was no significant difference in the expression of IκB at baseline, post-TNFα exposure or in the degree of IκB decrease following TNFα exposure between different melanoma genotypes (Figure 1B,C).

Transcriptome analysis was performed in five cutaneous (HT144, SKMel28, MM200, D22, NM182) and five uveal (MEL270, MP38, OMM1.3, MP46, MEL285) cell lines that were treated with exogenous TNFα for 24 h. Single sample gene set enrichment analysis (ssGSEA) confirmed the consistent and significant upregulation of the Hallmark_TNFA_signaling_via_NFKB gene signature (Figure 1D). We also noted that several other Hallmark gene signatures were upregulated in response to exogenous TNFα, including signatures of IL6_JAK_STAT3_Signaling, Inflammatory_Response, Allograft_Rejection, Interferon_Gamma_Response and Apoptosis gene signatures (Figure 1E). These data confirm that cutaneous and uveal melanoma cell lines consistently respond to TNFα stimulation.

### 3.2. TNFα-Induced Changes Are Variable and Reflect the Melanoma Differentiation States

One critical effect of TNFα is the induction of melanoma dedifferentiation [13], characterized by the downregulation of the melanocytic antigen Melan A (*MLANA* gene) and the microphthalmia-associated transcription factor MITF, and the upregulation of NGFR and the AXL receptor tyrosine kinase. The expression of these four markers has been used to subclassify melanoma into four progressive differentiation states, including undifferentiated, neural crest-like, transitory and melanocytic [21].

We examined TNFα effects on melanoma differentiation by analyzing the accumulation of the key marker proteins MITF, Melan A, AXL and NGFR in the 40 melanoma cell lines (Figure 2A and Appendix A). The majority of melanoma cell lines (23/40: 15/31 cutaneous, 8/9 uveal) showed features of the melanocytic state, including the high expression of MITF and Melan A along with low levels of AXL and/or NGFR. Another 14/40 melanoma cell lines (13/31 cutaneous and 1/9 uveal) displayed the neural crest-like dedifferentiated subtype with low MITF and Melan A, and elevated, albeit variable levels, of AXL and/or NGFR. Three cutaneous melanoma cell lines (C013M, MeWo and NM177) were classified as transitory, as they expressed elevated levels of MITF and Melan A, and intermediate/high levels of AXL and/or NGFR. Importantly, the classification of melanoma differentiation states was concordant between the gene expression and protein analyses (data not shown).

We next examined whether TNFα promoted melanoma state-specific changes. In melanocytic melanoma, TNFα was a potent suppressor of MITF and Melan A (Figure 2B,C). Unexpectedly, the downregulation of MITF was not associated with AXL upregulation (Figure 2D), and the expression of these two markers were not correlated in the melanocytic melanoma subtype (Figure 3A). NGFR was induced (fold change > 1.5) in 18/23 (78%) melanocytic cells and the melanocytic melanoma cells showing minimal or no TNFα-mediated NGFR induction (fold change < 1.5) displayed low NGFR expression at baseline (mean fluorescence intensity < 20) (Figure 2E). Although we did not perform statistical analyses on the three transitory melanoma cell lines, TNFα clearly reduced both MITF and Melan A levels and concurrently increased NGFR and AXL expression in this melanoma subtype. Finally, TNFα promoted variable responses in the neural crest-like dedifferentiated melanomas. MITF and Melan A expression were very low at baseline, and showed minimal, albeit consistently diminished expression in response to TNFα. The upregulation of AXL and NGFR expression was inconsistent and most dedifferentiated melanomas did not show substantial increases in AXL or NGFR in response to TNFα. In fact, AXL was induced (fold change > 1.5) in only 4/14 (29%; SCC14-0257, HT144, MelAT, D24M) while NGFR was induced (fold change > 1.5) in 4/14 (29%; NM16, MM200, HT144, D22) dedifferentiated melanoma cell lines (Figure 2A). Interestingly, MITF was negatively correlated with AXL post TNFα in the dedifferentiated melanoma subtype (Figure 3B).

Collectively, these data indicate that TNFα was a potent suppressor of the MITF transcription factor and the associated melanoma antigen Melan A, particularly in the transitory and melanocytic melanomas, whereas NGFR and AXL induction were variable. Further, the melanoma cell state-specific changes did not reflect differences in TNFα signaling, as determined by the loss of the IκB protein (Appendix A).

### 3.3. Dedifferentiation Induced by Short Term Exposure to TNFα Is Reversible

To examine whether protein changes induced in response to TNFα were reversible, melanoma cells were treated with 1000 U/mL TNFα or 0.1% BSA control for 72 h and the media was replenished with either fresh 1000 U/mL TNFα or 0.1% BSA for a further 96 h. TNFα effects were reversible in both melanocytic (SKMel28, MP46, MEL270, OMM1) and dedifferentiated (HT144, MM200) melanoma cell lines, with the partial to complete restoration of MITF and Melan A expression after TNFα removal (Appendix A). This was particularly interesting as, in our hands, long-term cultures of dedifferentiated melanoma cells derived from patient biopsies remain dedifferentiated in the absence of exogenous TNFα (data not shown).

### 3.4. Antigen Presentation Molecules Are Poorly Induced by TNFα in Melanoma

The loss of pigmentation antigens in response to TNFα is thought to contribute to the resistance to immune checkpoint inhibitors, and it is possible that this diminished antigen repertoire may be compensated for by the induction of antigen presentation molecules HLA-ABC and HLA-DR. However, we noted that the TNFα-mediated induction of HLA-ABC and HLA-DR was modest in melanoma (Figure 4A,B; only 5/40 melanoma cell lines showed induced HLA-DR cell surface expression (>1.5 fold induction; mean HLA-DR induction in 40 melanoma cell lines = 1.3) and although 24/40 melanoma cell lines showed some increase in HLA-ABC cell surface expression, the level of induction was less than two-fold in 12 of these 24 cell lines (mean HLA-ABC induction in 40 melanoma cell = 1.9). We have previously identified the loss of beta-2 microglobulin (B2M) in one cell line, SMU15-0217, which resulted in the absence of HLA-ABC [14]. The influence of TNFα on HLA-ABC and HLA-DR expression was similar across all melanoma differentiation states, and although HLA-ABC was only significantly induced in dedifferentiated melanoma cells, these cells did not express significantly more HLA-ABC or HLA-DR post TNFα treatment when compared to the melanocytic melanomas.

### 3.5. TNFα Promotes PD-L2 but Only Weakly Induces PD-L1 Expression in Melanoma

We next explored whether TNFα influences the expression of two key regulators of anti-tumor immunity, PD-L1 and PD-L2. Exogenous TNFα did not consistently induce the expression of PD-L1 (Figure 4C, only 3/40 melanoma cell lines, A2058, C086 and Mel270, showed >1.5 fold induction in response to TNFα). In contrast, TNFα induced PD-L2 in 21/40 melanoma cell lines, although the fold induction was inconsistent, ranging from 1.5 to 11.1, with a mean fold induction of 2.3. The upregulation of PD-L2 was independent of melanoma differentiation state; both melanocytic and dedifferentiated melanoma cells showed a similar degree of induction in response to TNFα (Figure 4D). PD-L1 and PD-L2 cell surface expression were significantly correlated with the *CD274* (PD-L1) and *PDCD1LG2* (PD-L2) transcript expression in the nine melanoma cell lines with matched transcriptome data (Appendix A). Finally, we noted that no single melanoma cell line showed the TNFα-mediated induction of the HLA-ABC, HLA-DR, PD-L1 and PD-L2 immune markers, even though TNFα signaling was intact in all melanoma cell lines tested.

## 4. Discussion

Our investigation of the effects of exogenous TNFα on a panel of 40 melanoma cell lines with different genotypes (i.e., BRAF-mutant, NRAS-mutant and BRAF/RAS wildtype), subtypes (i.e., cutaneous and uveal) and differentiation states (i.e., neural crest-like dedifferentiated, melanocytic and transitory) revealed that the TNFα signaling pathway is maintained in all melanomas. However, the downstream impact of TNFα is variable, likely due its complex and dynamic regulation of multiple signal transduction pathways.

TNFα is abundantly expressed by immune cells upon activation [3] and melanoma cells can respond to TNFα with dynamic phenotypic transitions [9,13,22]. We show that melanoma cells respond to TNFα by undergoing phenotypic switching in a cell state-specific manner, with melanocytic and transitory melanomas being more susceptible to TNFα-induced dedifferentiation; these cells showed pronounced downregulation of the MITF transcription factor and its downstream target Melan A, with concurrent upregulation of NGFR compared to the neural crest-like dedifferentiated melanomas. In contrast, the upregulation of AXL by TNFα was variable and only observed in 6/40 melanoma cell lines tested. The upregulation of AXL occurred predominantly in the transitory and dedifferentiated melanomas, suggesting that although MITF and AXL expression has been reported to be inversely correlated in melanoma [23], their regulation by TNFα appear dependent on variable mediators [24].

Consistent with previous reports [9,13], we show that the effects of TNFα on dedifferentiation are reversible; the removal of TNFα restored MITF and Melan A expression, although the expression of AXL remained unchanged. This is critical in the context of melanoma treatment, as dedifferentiation confers resistance to targeted therapies [23,25] and immunotherapies [13]. As TNFα is a key regulator of phenotypic plasticity in melanoma cells, it is tempting to speculate that the removal of TNFα or modulation of this pathway may divert melanoma cells from a dedifferentiated drug resistant state, and resensitize melanomas to therapies. Indeed, the blocking of TNFα has been suggested to boost the response to the immune checkpoint blockade, by enhancing tumor regression and preventing activation-induced CD8 T cell death in MC38 and B16-OVA mouse tumor models [26]. A phase 1b clinical trial (TICIMEL) is currently investigating the efficacy of infliximab or certolizumab (anti-TNFα) and immune checkpoint inhibitor combination in patients with advanced melanoma (NCT03293784). It is worth noting, however, that our dedifferentiated melanoma cells did not revert to their melanocytic state in the absence of TNFα, and this may reflect a temporal switch to an irreversible dedifferentiation state, that is driven by epigenetic changes [27].

In contrast to TNFα-induced dedifferentiation, the degree of the TNFα-mediated induction of HLA-ABC, HLA-DR, PD-L1 and PD-L2 was inconsistent and more difficult to predict. Induction was independent of melanoma genotype, subtype and differentiation state, although the induction of all four immune mediators was significantly correlated with baseline expression (data not shown). The TNFα-mediated induction of antigen presenting molecules was weak and/or uncommon; HLA-ABC showed a mean fold induction of only 1.9, while HLA-DR was induced in only 12.5% of melanoma cells, suggesting it to be a poor effector of tumor recognition. The cell surface expression of PD-L1 was also low in our panel of melanoma cells at baseline and after TNFα induction, although the PD-L1 transcript and protein are both significantly induced, and we have previously shown that this molecule is also highly inducible by IFNγ [14]. It is also important to note that no single melanoma cell line showed the TNFα-mediated induction of all four immune markers, further highlighting the variability in TNFα downstream signaling.

Our study shows that TNFα potently drives a program of melanoma dedifferentiation, particularly in melanocytic and transitory melanomas. TNFα-induced transcriptome and protein changes are also associated with a restricted and weak induction of antigen presentation molecules and the upregulation of the PD-L2 immune checkpoint. The combination of these dynamic changes supports a phenotype that is resistant to immunotherapies and provides markers that may be used to select patients for alternate trial therapies.

## Figures and Tables

**Figure 1 biomedicines-09-00605-f001:**
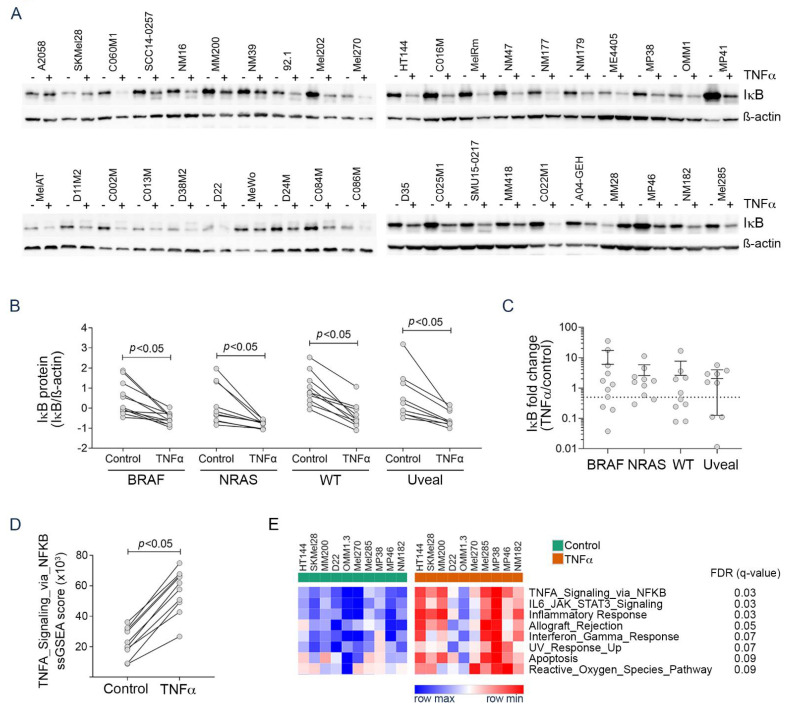
TNFα responses in cutaneous and uveal melanoma cell lines. (**A**) Western blots of melanoma cell lysates showing IκB expression 72 h after treatment with BSA control (-) or 1000 U/mL TNFα (+). (**B**) Densitometric quantitation of IκB expression (normalized to ß-actin and converted to z-scores to enable analysis of independent experiments) after treatment with BSA control or TNFα in BRAF-mutant, NRAS-mutant and wildtype (WT) cutaneous and uveal melanomas. Each dot represents the average of three independent experiments for each cell line. Significant comparisons between control and TNFα were determined by paired *t* test and indicated by *p* < 0.05. (**C**) Fold change in IκB expression (TNFα/control z-scores after ß-actin normalization). Each dot represents the average of three independent experiments for each cell line. Results are the average ± SD in each group and significant comparisons between groups determined by one-way ANOVA and Tukey’s multiple comparisons test. (**D**) Single sample gene set enrichment analysis (ssGSEA) scores for the Hallmark_TNFA_Signaling_via_NFKB in 10 melanoma cell lines (HT144, SKMel28, MM200, D22, NM182, MEL270, MP38, MP46, OMM1.3, MEL285) after BSA control or TNFα exposure. Comparisons between control and TNFα determined by paired *t* test and indicated by *p* < 0.05. (**E**) Heatmap showing differentially expressed Hallmark transcriptome gene sets between melanoma cell lines treated with BSA control or 1000 U/mL TNFα for 24 h. Differences were determined using a *t* test and significant signatures selected based on a false discovery rate (FDR) *q* < 0.10.

**Figure 2 biomedicines-09-00605-f002:**
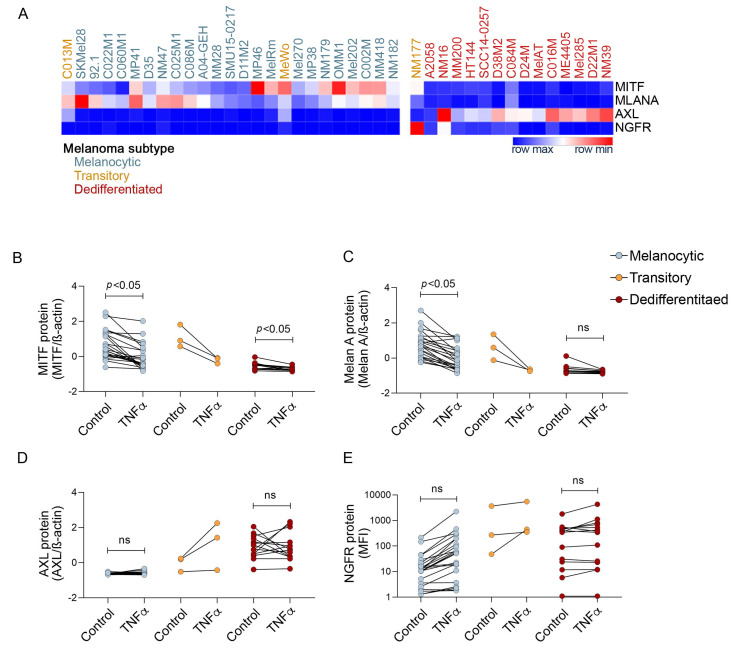
Expression of melanoma dedifferentiation markers in response to TNFα in 40 melanoma cell lines. (**A**) Heatmap showing the densitometric quantitation of MITF, Melan A and AXL (normalized to ß-actin and converted to z-scores), and geometric mean fluorescence intensity (MFI) values for NGFR in melanocytic (blue), transitory (orange) and dedifferentiated (red) melanoma cells at baseline (BSA control). The differentiation states of the melanoma cell lines were classified based on the mean z-scores of MITF, Melan A and AXL and the MFI for NGFR from three independent experiments. Relative expression of (**B**) MITF, (**C**) Melan A, and (**D**) AXL based on densitometric values, and (**E**) NGFR based on MFI values, 72 h after treatment with BSA control or TNFα. Each dot represents the average of three independent experiments for each cell line. Significant comparisons between groups were determined by paired *t* test and indicated by *p* < 0.05, ns; not significant.

**Figure 3 biomedicines-09-00605-f003:**
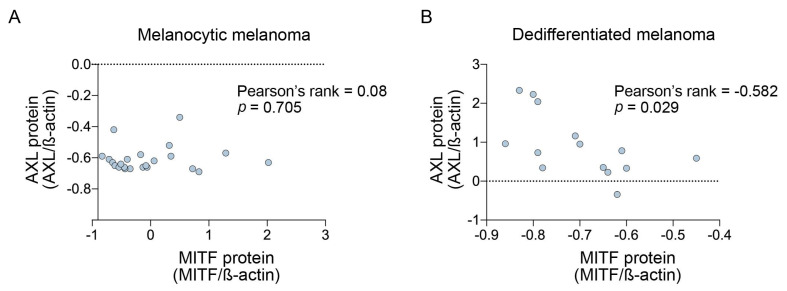
Correlation of melanoma dedifferentiation markers in melanocytic and dedifferentiated melanomas. Scatterplots showing correlation of AXL and MITF protein (normalized to ß-actin and converted to z-scores) in (**A**) melanocytic (*n* = 23) and (**B**) dedifferentiated (*n* = 14) melanoma cell lines. Correlation calculated using Pearson’s rank correlation coefficient, *p* < 0.05.

**Figure 4 biomedicines-09-00605-f004:**
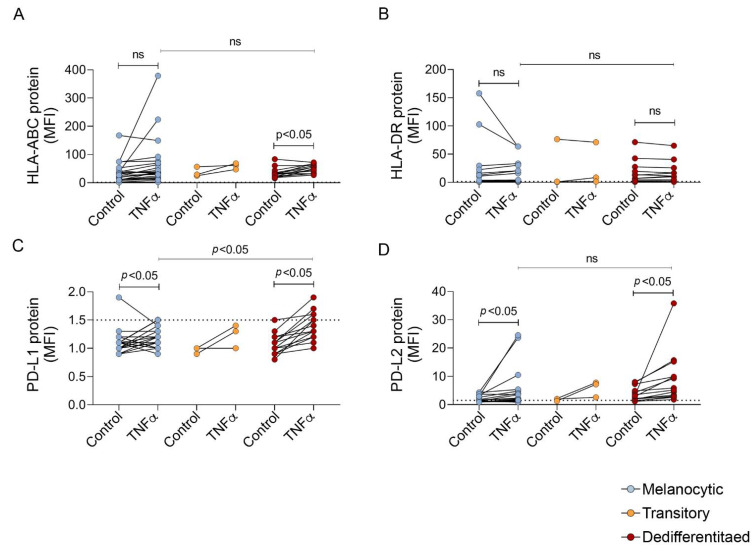
Induction of immune effector molecules by TNFα. Cell surface expression of (**A**) HLA-ABC, (**B**) HLA-DR, (**C**) PD-L1 and (**D**) PD-L2 (shown as relative mean fluorescence intensity (MFI)) 72 h after BSA control or TNFα treatment in melanocytic, transitory and neural crest-like dedifferentiated melanoma cell lines. Each dot represents one cell line and results are the average of at least three independent experiments. Dotted line shows a cutoff for positivity set at MFI = 1.5. Significant comparisons between groups were determined by paired (control vs. TNFα) or unpaired (TNFα vs. TNFα) *t* test and indicated by *p* < 0.05, ns; not significant.

## Data Availability

The data presented in this study are available on request from the corresponding author.

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
