# Peer review of "Melanoma Cell State-Specific Responses to TNFα"

_biomedicines, 2021, doi:10.3390/biomedicines9060605_

Round 1

Reviewer 1 Report

With a great interest I read the research paper “Melanoma cell state-specific responses to TNFα” by Lim at al. investigating the role of TNFa in diverse cutaneous and uveal melanoma cell lines’ phenotypic plasticity. Understanding the mechanism modulating loss of differentiation in patients with advanced melanoma treated with checkpoint inhibitors would be vital for providing a lower resistance, which is already investigated in TICIMEL trial. The paper by  Lim at al. shed new light on the relation of AXL and MITF expression in regard to TNFa stimulation and mechanisms arresting the cells in dedifferentiated state. The authors also suggest that the inverse correlation between AXL and MITF is lost in some melanoma cell lines treated with TNFa.

I have only minor technical suggestions in regard to the paper:
1. Please provide the uniformity of presenting the company (country of origin – as the style varies in the text). Same applies to z-score and p value (sometimes mentioned as “z score”; and P, respectively) in text and figure legends.
2. Please use provide the full term for mean fluorescence intensity in each legend, as every figure should be autonomous.

Author Response

We thank the reviewer for the kind comments and critique and have specifically addressed the reviewer's suggestions as detailed below.

1. Please provide the uniformity of presenting the company (country of origin – as the style varies in the text). Same applies to z-score and p value (sometimes mentioned as “z score”; and P, respectively) in text and figure legends.

We have now edited our manuscript to ensure uniformity in text throughout, including the source of consumables (country of origin), the z-score and p value. 

2. Please use provide the full term for mean fluorescence intensity in each legend, as every figure should be autonomous.

We have also provided the full term for mean fluorescence intensity (MFI) in each of the figure legends.

We have that the manuscript is now acceptable for publication.

Reviewer 2 Report

A report by Lim et al. aims at investigating the role TNFa on melanoma cells as a surrogate of microenvironment accompanying response to immunotherapy. This study, although lacks substantial novelty, presents technically valid data. The strength of the study is the use of multiple melanoma cell lines of different genetic subtypes. The conslusions are supported by the results.

The manuscript requires only unified editing service. In addition, the quality of the figures should be improved. 

Author Response

We thank the reviewer for the kind comments and review. We have addressed the reviewer's comments as detailed below:

1. The manuscript requires only unified editing service. In addition, the quality of the figures should be improved. 

We have now edited our manuscript to ensure uniformity in text and have included figures with better resolution and clarity.

We hope that the manuscript is now suitable for publication.